# Opposing effects of HNP1 (α-defensin-1) on plasma cholesterol and atherogenesis

Mohamed Higazi[1], Suhair Abdeen[1], Rami Abu-Fanne[1], Samuel N. Heyman[2], Aseel Masarwy[1], Khalil Bdeir[3], Emad Maraga[1], Douglas B. Cines[3], Abd Al-Roof Higazi[1,3]*

1 Department of Clinical Biochemistry, Hadassah-Hebrew University, Jerusalem, Israel, 2 Department of Medicine, Hadassah University Hospital, Mt. Scopus, Jerusalem, Israel, 3 Departments of Pathology and Laboratory Medicine, University of Pennsylvania-Perelman School of Medicine, Philadelphia, Pennsylvania, United States of America

* abdh@ekmd.huji.ac.il

**Data Availability Statement:** All relevant data are within the manuscript and its Supporting Information files. As requested by your journal: Anesthesia and euthanasia information were

## Abstract

Atherosclerosis, the predominant cause of death in well-resourced countries, may develop in the presence of plasma lipid levels within the normal range. Inflammation may contribute to lesion development in these individuals, but the underlying mechanisms are not well understood. Transgenic mice expressing α-def-1 released from activated neutrophils develop larger lipid and macrophage-rich lesions in the proximal aortae notwithstanding hypocholesterolemia caused by accelerated clearance of α-def-1/low-density lipoprotein (LDL) complexes from the plasma. The phenotype does not develop when the release of α-def-1 is prevented with colchicine. However, ApoE$^{-/-}$ mice crossed with α-def-1 mice or given exogenous α-def-1 develop smaller aortic lesions associated with reduced plasma cholesterol, suggesting a protective effect of accelerated LDL clearance. Experiments were performed to address this seeming paradox and to determine if α-def-1 might provide a means to lower cholesterol and thereby attenuate atherogenesis. We confirmed that exposing ApoE$^{-/-}$ mice to α-def-1 lowers total plasma cholesterol and decreases lesion size. However, lesion size was larger than in mice with total plasma cholesterol lowered to the same extent by inhibiting its adsorption or by ingesting a low-fat diet. Furthermore, α-def-1 levels correlated independently with lesion size in ApoE$^{-/-}$ mice. These studies show that α-def-1 has competing effects on atherogenesis. Although α-def-1 accelerates LDL clearance from plasma, it also stimulates deposition and retention of LDL in the vasculature, which may contribute to development of atherosclerosis in individuals with normal or even low plasma levels of cholesterol. Inhibiting α-def-1 may attenuate the impact of chronic inflammation on atherosclerotic vascular disease.

## Introduction

Atherosclerosis and its thrombotic sequelae are the predominant causes of death in highly resourced countries and its incidence is increasing in "developing" parts of the world[1].

included in the method section. The methodology section regarding the quantification of lipid deposition was expanded. Photographic evidence supporting the intimal damage score results shown in Fig 1B were added in the new Fig 1C.

**Funding:** This work was supported by HL123912 (AAH), HL139448 (DBC) and a grant from the Israeli Science Foundation (AAH).

**Competing interests:** The authors have declared that no competing interests exist.

Hypercholesterolemia is a well-established risk factor for atherosclerosis and lowering plasma levels of LDL-cholesterol through a variety of approaches mitigates risk[2]. However, between 15 and 50% of all cardiovascular events occur in patients with plasma cholesterol within the normal range[1,3]. The mechanisms that drive development of atherosclerosis in such individuals are not well established.

There is compelling data that point to the contribution of inflammation to the development of atherosclerotic lesions[3–7]. More specifically, results from several groups using diverse approaches suggest that human α-defensins (α-defs) 1–4, also known as human neutrophil peptides (HNPs),[2] contribute to this process[5,6,8–11]. We recently examined the mechanism by which α-defensins released from activated neutrophils may promote formation of lipid streaks in the vasculature using HNP-1tg/tg Def$^{+/+}$ (Def$^{+/+}$) mice fed a high fat diet (HFD)[12]. We found that α-defensin-1 (α-def-1) forms complexes with LDL, accelerating lipoprotein clearance from the circulation by the liver, leading to hypocholesterolemia[12]. In line with this, plasma levels of total cholesterol (TCH) and LDL were lower in Def$^{+/+}$ mice than in wild type control animals[12]. However, the Def$^{+/+}$ mice developed more prominent lipid streaks and greater monocyte/macrophage retention and greater generation of cathepsins B and S in the proximal aorta compared with wild type controls, consistent with enhanced retention of LDL/α-def-1 complexes in the vasculature[12]. Phenotypic correction was seen when release of α-def-1 by neutrophils was blocked by colchicine or precluded by bone marrow transplantation from wild type mice[12].

More recently, formation of LDL/ α-def-1 complexes, enhanced blood clearance of LDL and low plasma LDL were also found in mice double transgenic for α-def-1 and apoE (ApoE$^{-/-}$HNP-1tg/tg) and in ApoE$^{-/-}$ mice given exogenous α-def-1[13]. However, in this context, the lower plasma levels of LDL were associated with attenuated development of lipid streaks, showing an apparent protective effect of α-defs in this model and suggesting these peptides might serve as a pharmacological vehicle to prevent atherosclerosis[13].

Here, we asked whether the development of lipid streaks in this latter model reflects a balance between the lowering of plasma cholesterol through enhanced hepatic clearance offset by increased binding and retention of residual circulating LDL/α-def-1 complexes in the vasculature[5,12,14–17]. If so, we reasoned that the vascular lesions would be more prominent in ApoE$^{-/-}$HNP-1tg/tg, or ApoE$^{-/-}$ mice exposed to exogenous α-def-1 than if LDL were lowered to the same extent through alternative mechanisms. We found that using cholestyramine or diet modification to lower plasma cholesterol and LDL in ApoE$^{-/-}$ mice to the levels seen in ApoE$^{-/-}$HNP-1tg/tg leads to smaller lipid streaks than those developed in ApoE$^{-/-}$ mice exposed to exogenous α-def-1. The implications of our findings regarding the pathogenesis of atherosclerosis and its prevention are discussed.

## Methods

### Materials

Regular rodent chow diet (RD; 6.5% fat) and HFD (15.8% fat and 1.25% cholesterol) (TD.88051, Harlan) were purchased from Harlan (Harlan, Rehovot Israel) and cholestyramine from Sigma-Aldrich. α-def-1 was purchased from Sigma-Aldrich and kindly provided by Dr. Wuyan Lu (Univ. MD School of Medicine). Each formed complexes with LDL *in vitro* and accelerated their clearance *in vivo*[12] and were used interchangeably.

### Mice

ApoE$^{-/-}$ mice on a C57BL/6 background were bred in-house from a stock originating from Jackson Laboratories provided by M. Aviram (Rappaport Faculty of Medicine, Technion,

Haifa, Israel). Animal care and experiments were conducted in accordance with protocols approved by the Animal Care Committee of the Hebrew University (approval number: MD-15-14579-4) and the University of Pennsylvania. Mice were maintained on a regular rodent chow diet, on an HFD, or a moderate high fat diet (MFD) by combining 65% RD with 35% HFD for the indicated times.

### Cholesterol-lowering

Five approaches were used to modify plasma cholesterol in ApoE$^{-/-}$ female mice (16 per group). One set of mice was fed a HFD for 6 weeks without or with 1.5% or 3% cholestyramine [18]. A second set of mice on a HFD, received an intravenous (IV) injection, via tail vein, of α-def-1 (10 or 30 μg) or vehicle (PBS) control every other day[13,19]. A sixth group, studied in parallel, was fed a MFD for 6 weeks. Mice were monitored for adverse effects throughout the experiment. There were no statistically significant differences in body weight between mice in any of the six groups. Mice were anesthetized with an intraperitoneal injection of zolazepam (25 mg/kg) and xylazine (50 mg/kg) on the last day of the experiment and blood samples were taken by transcardiac puncture after 6 hours of fasting[12]. Serum total cholesterol (TCH) and high-density lipoprotein cholesterol (HDL) were measured by enzymatic methods using an autoanalyser (Cobas 6000; Roche, Nakakojo, Japan), and levels of low-density lipoprotein cholesterol (LDL) were calculated as reported[12].

### Staining of aortic roots

After blood was withdrawn, mice were euthanized with pentobarbital and the hearts were immediately removed and transected midway between the apex and base in a plane parallel to a line defined by the tips of the atrial appendages. The basal ventricular segments in continuity with the atria and aortic roots were embedded in optimum cutting temperature compound (OCT) and frozen in liquid nitrogen. Cryostat sections were prepared at ~7 μm intervals using a CM 1900 cryotome (Leica Microsystems, Wetzlar, Germany), fixed in formalin, stained with Oil Red, and examined to assess proximity to the aortic root. Sections through the coronary ostia, coronary sinuses, and the aortic leaflets were captured. Lipid deposition was quantified in parallel sections stained with Oil Red, and quantified using Image-Pro Plus analysis software as previously reported[12]. Results are reported as the percentage of the circumference of each root that was Oil Red positive.

### Statistical analysis

Group comparisons were performed using one-way ANOVA with the Newman-Keuls post hoc test[12,20]. Correlations between plasma cholesterol and the lesion score were calculated for the entire cohort and separately for each experimental group. Multivariate regression analysis was applied to assess the independent impact of α-def-1 on the size of fatty streaks, irrespective to cholesterol levels. Data is presented as means ± SD, and statistical significance was set at $p < 0.05$.

### Results

We examined the effect of exogenous α-def-1 on total plasma cholesterol (TCH) and the extent of fatty streaks formed in the aortic roots of ApoE$^{-/-}$ mice fed a HFD. In the first set of experiments, mice were given 10 or 30 μg α-def-1 IV (low dose and high dose, respectively) or saline vehicle, every other day. Injection of low dose and high dose α-def-1 decreased serum TCH from 1425.4 ± 177.3 to 976.5 ± 160.7 and 707.6 ± 65.3 (mg/dl), respectively ($p < 0.001$ vs.

untreated mice) (Fig 1A), accompanied by proportional decrease in LDL, confirming previous reports from our group[12] and others[13]. The percentage of the circumferences of aortic roots occupied by fatty streaks decreased from 39.9 ± 6.6 to 34.4 ± 3.7% (p < 0.05) and 23.4 ± 2.5% (P< 0.001), respectively following injection of α-def-1 (Fig 1B & 1C), both consistent with previous findings[13].

We then compared the effect of α-def-1 with that of cholestyramine. TCH in ApoE$^{-/-}$ mice on HFD given a low (1.5%) or high (3%) dose of cholestyramine fell to 1108.9 ± 112.9 and 756.1 ± 62.6 mg/dl (p < 0.001 vs. untreated mice), respectively (Fig 1A), and the percentage of the aortic root circumferences occupied by fatty streaks decreased from 39.9 ± 6.6 to 20.5 ± 2.5% and14.1 ± 2.3%, respectively (p < 0.001 compared with untreated mice) (Fig 1B & 1C), also consistent with previously reported findings[18].

Of note, however, lesion sizes in mice given high dose α-def-1 were significantly larger than those in mice given the high dose of cholestyramine (p< 0.001) (Figs 1B & 1C & 2), although the two experimental groups had almost identical average plasma levels of TCH (Figs 1A & 2). Lesion size was also smaller in mice fed a modified high fat diet (MFD) (Figs 1B, 1C & 2) that had plasma TCH levels comparable to those observed in the high-dose α-def-1 group (731.9 ± 49.36 mg/dl) (Figs 1A & 2). Lesion size in MFD mice group was comparable to HFD mice given the higher dose of cholestyramine (Figs 1B, 1C & 2), with both groups having equivalent levels of plasma TCH (Figs 1A & 2).

To isolate the atherogenic effect of α-def-1 in ApoE$^{-/-}$ mice, we examined the relationship between plasma cholesterol and lesion size in presence and absence of α-def-1. A close correlation between serum cholesterol and lesion size was observed in all 6 cohorts combined (Fig 2) (R = 0.70, p<0.0001). However, a significantly higher proportion of the aortic roots circumferences were occupied by lesions in mice given low and high doses of α-def-1 (Fig 2, results above the regression line) compared with the other groups of mice (positioned below the regression line) having comparable levels of TCH (p<0.0001; Fig 2). These results strongly suggest an independent effect of α-def-1 on lesion size.

As a second approach to evaluate the potential atherogenic effect of α-def-1, we used multiple regression analysis with the sizes of fatty streaks defined as the dependent variable and cholesterol and α-def-1 serving as stepped predictors. As expected, the impact of plasma cholesterol alone was substantial (Beta 0.867, F-to-remove 287.9, p<0.0001). However, α-def-1 continued to show an independent effect on lesion size (Beta 0.565, F-to-remove 122.3. p<0.0001), consistent with previously reported findings[5].

## Discussion

We previously reported data suggesting that α-defensins promote atherogenesis, based on studies in transgenic mice expressing α-def-1 in their neutrophils[12] (Def$^{++}$ mice) that have been fed a high fat diet[12] and by the correlation in humans between tissue deposition of α-defensins and the severity of coronary artery disease[5]. The increase in the size of lipid rich lesions in the proximal aortas of Def$^{++}$ mice[12] fed a HFD was observed even though α-def-1 stimulated hepatic clearance of LDL leading to hypocholesterolemia[12]. However, a decrease in lesion size was seen when ApoE$^{-/-}$ mice were crossed with the same Def$^{++}$ mice or were exposed to exogenous α-def-1 showing an apparent protective effect, presumably as a result of accelerated LDL clearance from the plasma[13].

To address these seemingly conflicting observations, the current study was designed to isolate the independent impact of α-def-1 on atherogenesis using control groups with comparable lipid profiles attained through other approaches. We extended previously published reports by including two different concentrations of α-def-1 and three control groups, i.e. two groups of

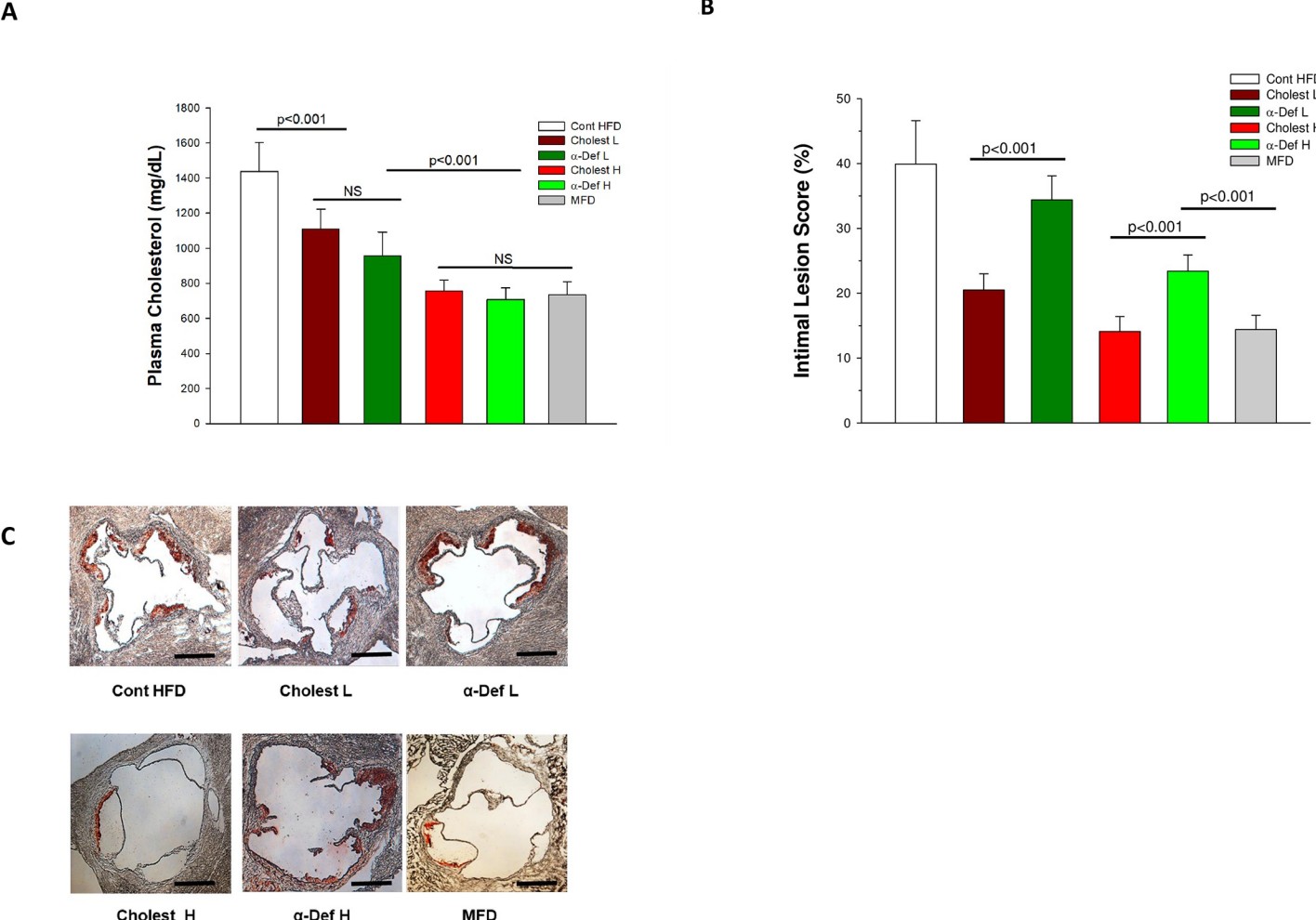

**Fig 1. Effect of intervention on plasma cholesterol and development of lipid streaks in aortic roots. Panel A. Cholesterol levels**. ApoE$^{-/-}$ mice fed a HFD for 6 weeks were divided into 5 groups (n = 16/group). One went untreated (Cont). Two groups were given two oral doses of either a low or high dose cholestyramine, 1.5% (Choles L) or 3% (Choles H), respectively. Two groups were given two IV injections of α-def-1, 10 or 30 μg, every other day (α-def-1 L) and (α-def-1 H) respectively. A sixth group was given a modified high fat diet (MFD) for 6 weeks. Blood samples were taken after 6 hours of fasting. Plasma levels of total cholesterol (TCH) were measured. **Panel B. Development of lipid streaks.** ApoE$^{-/-}$ mice on HFD or MFD for 6 weeks described in Panel A were sacrificed after blood draw. Lesion size in sections from the aortic roots (See Panel C) was measured as described in the Methods section. Results are expressed as the percent of the aortic root circumferences stained by Oil Red O. The mean ± S.D. and p < 0.05 values are shown; n = 16 per group. **Panel C. Lipid streaks.** Representative sections from the various experimental groups are shown. Scale bars represent 500μm.

mice given high or low doses of cholestyramine and a cohort of mice fed a MFD. Our data show that despite a comparable decline in plasma cholesterol, mice exposed to α-def-1 developed larger aortic lesions. This suggests that the lipid-lowering impact of α-def-1 might be offset by induction of a pro-atherogenic processes within the vasculature itself, leading to the net effect of accelerating the generation of intimal fatty streaks.

Our results confirm previous finding that α-def-1, here given exogenously to ApoE$^{-/-}$ mice, lowers total serum cholesterol and decreases lesion size in the aortic root[13]. Similar effects were seen when plasma cholesterol was lowered by inhibiting adsorption using cholestyramine or using a MFD. However, although total serum cholesterol was lowered to the same extent by both methods, lesion size was larger in mice given α-def-1 (Fig 1) and there was a statistically

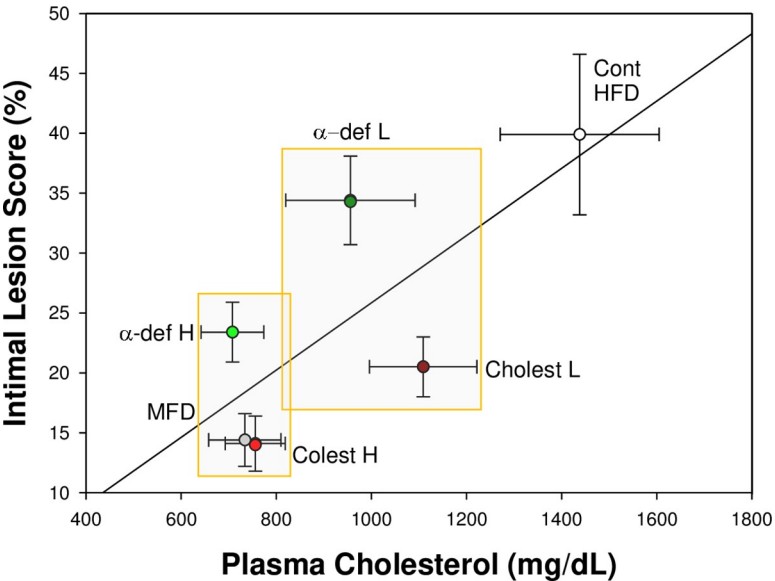

**Fig 2. Correlation between plasma cholesterol and size of lipid-rich streaks in aortic roots.** Data from the six groups of ApoE$^{-/-}$ mice described in Fig 1, panels A and B are compared (regression line), and separately for each experimental group (mean ± SD, n = 16 per group). A highly significant correlation is noted in between plasma cholesterol and lesion size, plotted for the entire cohort of mice (R = 0.7, p<0.0001, Pearson's correlation). The two rectangles illustrate experimental groups with comparable plasma cholesterol but significantly higher lesion size score in the α-def-1 groups (p<0.001, ANOVA).

significant direct correlation between exposure to α-def-1 and lesion size that was independent of plasma cholesterol (Fig 2).

Similar conclusions can be reached from the data of Paulin[13] et al. ApoE$^{-/-}$ mice that had been fed a HFD and were also exposed to exogenous α-def-1 that had serum TCH levels between 1000–1500 mg/dl developed larger lesions than untreated mice with comparable serum levels (see Fig S3D[13]).

Taken together, our results suggest that α-def-1 has competing effects on atherogenesis. On the one hand, accelerated hepatic uptake and clearance of α-def-1/LDL complexes from the blood reduces their probabilities to deposit in the vasculature. On the other hand, α-def-1/LDL complexes have a greater intrinsic propensity to deposit and remain in vascular cells and matrix[5,12,14–17], where they stimulate macrophage recruitment, generation of cathepsins B and S, formation of foam cells[10] and increase endothelial permeability to LDL[12]. The seemingly paradoxical inverse relationship between low plasma LDL levels and lipid deposition in vasculature induced by α-def-1 that we observed helps to explain recent findings that colchicine, which reduces α-defensin release from neutrophils[12], increases plasma oxidized LDL and proatherogenic "small" LDL[21] but reduces the incidence of cardiovascular events [22]. It is clear that more research will be needed to fully evaluate the impact of α-defensins on atherogenesis and to explore the possibility that inhibiting their release and impact on lipid deposition might provide a novel approach to the amelioration of atherogenesis.

## Supporting information

**S1 Data.**
(DOCX)

## Author Contributions

**Conceptualization:** Mohamed Higazi, Abd Al-Roof Higazi.

**Data curation:** Mohamed Higazi, Suhair Abdeen, Rami Abu-Fanne, Aseel Masarwy, Emad Maraga.

**Formal analysis:** Mohamed Higazi, Rami Abu-Fanne, Samuel N. Heyman, Khalil Bdeir, Douglas B. Cines, Abd Al-Roof Higazi.

**Investigation:** Suhair Abdeen, Aseel Masarwy, Emad Maraga, Douglas B. Cines.

**Supervision:** Abd Al-Roof Higazi.

**Validation:** Suhair Abdeen, Aseel Masarwy, Emad Maraga.

**Visualization:** Khalil Bdeir, Abd Al-Roof Higazi.

**Writing – original draft:** Mohamed Higazi, Douglas B. Cines, Abd Al-Roof Higazi.

**Writing – review & editing:** Samuel N. Heyman, Douglas B. Cines, Abd Al-Roof Higazi.

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
