## [Decision Letter · Decision Letter 0]

30 Jan 2020

PONE-D-19-31479

Opposing effects of HNP1 (α-defensin-1) on plasma cholesterol and atherogenesis

PLOS ONE

Dear Prof. Higazi,

Thank you for submitting your manuscript to PLOS ONE. After careful consideration, we feel that it has merit but does not fully meet PLOS ONE’s publication criteria as it currently stands. Therefore, we invite you to submit a revised version of the manuscript that addresses the points raised during the review process.

We would appreciate receiving your revised manuscript by Mar 15 2020 11:59PM. To enhance the reproducibility of your results, we recommend that if applicable you deposit your laboratory protocols in protocols.io, where a protocol can be assigned its own identifier (DOI) such that it can be cited independently in the future. For instructions see: http://journals.plos.org/plosone/s/submission-guidelines#loc-laboratory-protocols

We look forward to receiving your revised manuscript.

Kind regards,

Michael Bader

Academic Editor

PLOS ONE

2. Please expand the acronym “HL123912 and HL139448” (as indicated in your financial disclosure) so that it states the name of your funders in full.

Reviewers' comments:

Reviewer's Responses to Questions

**Comments to the Author**

1. Is the manuscript technically sound, and do the data support the conclusions?

Reviewer #1: No

Reviewer #2: Partly

2. Has the statistical analysis been performed appropriately and rigorously? 

Reviewer #1: I Don't Know

Reviewer #2: Yes

3. Have the authors made all data underlying the findings in their manuscript fully available?

Reviewer #1: Yes

Reviewer #2: Yes

4. Is the manuscript presented in an intelligible fashion and written in standard English?

Reviewer #1: No

Reviewer #2: No

5. Review Comments to the Author

Reviewer #1: The authors re-examine the potential role of HNP in atherosclerosis. It seems that this study is a reassessment of their own work and one other study, namely ref. 13; it is not very clear to this reviewer what the rationale behind this study is and what the added value is supposed to be.

Specific:

This study is hard to read with many typos and syntax errors. This manuscript needs much rewriting.

Basic standards in reporting are not adhered to, e.g. what is a one-way test? scale bars in images are missing; source of reagents (HNP) are missing; MFD seems like a random mix of different foods rather than a standardized diet, def-low group is shown twice in 1C, but def-hi is missing - these are just some examples but there are certainly many more.

The authors correlate plasma cholesterol levels with lesions sizes across different treatment groups and draw conclusions as to the mechanism of HNP; I am not certain this is sound as atherosclerosis develops over a long period and assessing cholesterol at just one time point likely does not reflect the timely resolution that is needed to draw the conclusions as they do.

In 1B they measure lesion size (please do not call this damage score!) and the control group is meant to range at 40%; however, looking at images in 1C, lesion size looks more like 6-10%; hence, I much question the overall robustness of their readout.

It is very unusual to see the work of another group being reinterpreted even with making reference to a supplementary figure.

Reviewer #2: Note for editor: are figure panels required to be separated before combining into one figure?

In this study Higazi et al. demonstrate in ApoE-/- mice fed a high fat diet (HFD), administration of human alpha-defensin 1 reduces plasma total cholesterol as well as lesion volume in comparison to control HFD untreated mice. In contrast to low vs high treatment with cholestyramine, the high and low alpha-def-1 treatment had larger lesion sizes, however this is still smaller than mice not receiving any alpha-def-1. The authors present this study in order to resolve the differences between their previously publication, where expression of defensins in a HFD model developed lipid streaks but formed adef:LDL complexes and the study by Paulin et al. wherein defensin expression in a HFD lowered progression of atheroprogression, enhanced LDL clearance and lowered cholesterol.

Throughout the manuscript, there are spelling errors to double check as well as punctuation, these need addressing.

In alpha-def-1 IV injection methodology, was 10 or 30 µg the total amount or is this µg/kg of bodyweight? This is human alpha-def-1? How much did the overall weight of the mice vary to account for the differences in relative amount of alpha-def-1 received by each individual across the treatment cohort. In addition, it is unclear if this is a tail vein injection.

Def+/+ mice need to be defined when introduced in the discussion, and rationale for using ApoE-/- mice be made more clear.

Figure legends.

1A. Clarify H & L indicate high and low, this can be included in the key as there is space. Whether the last gray bar is MFD or MHFD needs to be resolved, this and the high/low can be applied to subsequent panels. Finally, what is not significant and what is significant as indicated by the p<0.001 in the middle of the figure?

1B. The red and green asterisk description at present is unclear whether there is significant differences between low does Cholest or alpha-def 1 or whether the high and low dose treatments were compared. The Y axis labelling needs to be more clearly described in the legend.

1C. Figure legend needs to include this is representative aortic root sections... Also, there is no alpha-def high representative image.

2. Is intimal damage score expressed as a percentage similar to Figure 1B. At present, the legend is difficult for the reader to interpret that data from Figure 1 is being analyzed and needs clarification.

Overall, significant clarification in the discussion is needed. Clearly, there are multiple variables at play between alpha-def-1 and atheroprogression, especially in previous studies and the data presented in the current study, however the conclusions need to better match the data presented herein and outline what additional questions remain. First, this study attempts to control for the cholesterol-lowering effects of alpha-def-1 by including a treatment group is administered cholestyramine, and extends on previously published reports by including two different concentrations of alpha-def-1. It needs be abundantly clear in the text that while similar decreases in cholesterol were observed, alpha-def-1 treated mice had larger lesion size in comparison to cholestyramine treated mice. At present, what the authors are communicating by including the second to last paragraph of the discussion is unknown, however the correlation between the two studies overall shows a correlation between reductions in cholesterol and lesion size with the presence of alpha-def-1.

There are several key questions, would combination of cholestyramine and alpha-def-1 increase the lesion volume (due to alpha-def-1 activities?)? If alpha-def-1 was inhibited (or inactivated) in the present study, would that serve to increase both lesion size and cholesterol levels, or would lesion size be reduced in comparison to the ApoE-/- mice on the HFD, both in the presence or absence of cholestyramine? Would genetic modification to have Def+/+/ApoE-/- mice on a HFD result in mice with more atherogenesis that is reduced with something like colchicine, all these data would be helpful in supporting the authors current claims.

The data and previous studies show clearly that alpha-def-1 and neutrophils play a role in atherogenesis, one that might be a dual role, however the final sentence stating ‘inhibition of alpha-defs release or activity might help reduce the incidence or severity of atherosclerotic vascular disease and its thrombotic sequelae’ doesn’t appear to be supported currently by the data presented by the authors as in comparison to the control HFD, cholesterol and lesions are reduced.

6. PLOS authors have the option to publish the peer review history of their article (what does this mean?). If published, this will include your full peer review and any attached files.

Reviewer #1: No

Reviewer #2: No

---

## [Author Response · Author response to Decision Letter 0]

12 Mar 2020

Reviewer #1: The authors re-examine the potential role of HNP in atherosclerosis. It seems that this study is a reassessment of their own work and one other study, namely ref. 13; it is not very clear to this reviewer what the rationale behind this study is and what the added value is supposed to be.

We appreciate the opportunity to address the Reviewer’s concern, which is central to the merit of our paper. The reasons for submitting this paper are several fold. First, as stated succinctly by Reviewer 2, there are two peer-reviewed published manuscripts that address the role of human neutrophil alpha defensins (HNPs) in the development of atherosclerosis, one indicating these peptides are “proatherogenic”, the other concluding that these peptides inhibit the process and indeed could be employed to prevent or to reverse development of vascular lesions. We think the current paper helps to put both sets of data and both conclusions in a broader content, leading to a more nuanced conclusion that is more likely to provide insights into the human condition. Second, we also think this study provides insight into the pathogenesis of atherosclerosis in the sizable subpopulation of patients with normal or low levels of LDL for whom inhibition of neutrophil function is more likely to be salutary that focusing exclusively on lowering LDL cholesterol further. This latter hypothesis is supported by our finding that mice transgenic for alpha-defensin-1 (�-def-1) develop aortic plaques on a regular diet in the context of lower than normal plasma levels of LDL cholesterol and that colchicine prevents this from happening even when LDL cholesterol levels in the plasma are elevated by a high fat diet (J Biol Chem. 2016; 29: 2777). Our conclusions are supported by recent clinical studies that show colchicine prevents cardiovascular events in humans (N Engl J Med. 2019; 381: 2497) while increasing in the plasma, the more atherogenic LDL subspecies, small LDL and oxLDL (J Clin Lipidol. 2019; 13: 1016). 

Specific:

This study is hard to read with many typos and syntax errors. This manuscript needs much rewriting.

We apologize to the Reviewer and have made a sincere effort to avoid such errors in the revised text.

Basic standards in reporting are not adhered to, e.g. what is a one-way test? 

One-way ANOVA means performing between-group comparisons or within-group comparisons of repeated measurements, but not their combination (an assessment termed two-way ANOVA). Thank you for your remark. In the revised version we substituted "one-way ANOVA" with "between-group ANOVA" for simplicity. 

scale bars in images are missing; 

We have added the scale bars per the Reviewer.

source of reagents (HNP) are missing; 

We have added details concerning the sources of HNPs per the Reviewer (Please page 4, Materials section lines 3- 5).

MFD seems like a random mix of different foods rather than a standardized diet, 

We appreciate the Reviewer’s concern and we apologize for any confusion that we created. The diet we used was actually tightly controlled. The diet was composed of 35 parts high fat diet (See attached) and 65 parts regular diet (see attached), a conventional approach used by commercial companies to prepare specific diets. This composition was chosen based on empirical evidence that it generated plasma levels of total cholesterol and LDL cholesterol comparable to plasma levels in animals treated with low dose of cholestyramine and because it led to formation of lipid streaks in the aortas of comparable size. We used this as a second independent control to animals exposed to a low dose of HNPs. 

def-low group is shown twice in 1C, but def-hi is missing – 

 Thank you for noticing the mistake, we have corrected the error in labeling.

The authors correlate plasma cholesterol levels with lesions sizes across different treatment groups and draw conclusions as to the mechanism of HNP; I am not certain this is sound as atherosclerosis develops over a long period and assessing cholesterol at just one-time point likely does not reflect the timely resolution that is needed to draw the conclusions as they do.

We appreciate and, of course, agree with the Reviewer’s point as to the duration required to develop vascular disease. In experiments not shown, we measured plasma LDL and cholesterol weekly. Differences between Def++ and WT mice were already evident by the end of the first week and these differences remained constant throughout the experimental period. The data shown in this paper is representative of many such determinations, all leading to the same conclusion. 

In 1B they measure lesion size (please do not call this damage score!) and the control group is meant to range at 40%; however, looking at images in 1C, lesion size looks more like 6-10%; hence, I much question the overall robustness of their readout.

The Reviewer is correct. We expressed our data as the % of aortic root circumference occupied by the lesion. We provide a figure typical of those that we used to make these calculations. The data in the manuscript are presented as the ratio between the length of the red color line and the blue color line (See left figure below), which correlates very well with the size of the lesion (See right figures below). The Methods section of the paper has been amended (Please see page 5, line 6-7) and we changed the label of the Y axis in the figures 1B and 2 from "lesion size" to “damage score".

It is very unusual to see the work of another group being reinterpreted even with making reference to a supplementary figure.

We appreciate and concur with the Reviewer’s concern. We simply could not figure out how NHPs could be both pro-atherogenic and anti-atherogenic. We trust that we have been scrupulously fair to Paulin and colleagues and that our new findings help to reconcile what otherwise would appear to be opposing conclusions.

Reviewer #2: Note for editor: are figure panels required to be separated before combining into one figure?

In this study Higazi et al. demonstrate in ApoE-/- mice fed a high fat diet (HFD), administration of human alpha-defensin 1 reduces plasma total cholesterol as well as lesion volume in comparison to control HFD untreated mice. In contrast to low vs high treatment with cholestyramine, the high and low alpha-def-1 treatment had larger lesion sizes, however this is still smaller than mice not receiving any alpha-def-1. The authors present this study in order to resolve the differences between their previously publication, where expression of defensins in a HFD model developed lipid streaks but formed adef:LDL complexes and the study by Paulin et al. wherein defensin expression in a HFD lowered progression of atheroprogression, enhanced LDL clearance and lowered cholesterol.

Throughout the manuscript, there are spelling errors to double check as well as punctuation, these need addressing.

We apologize to the Reviewer. We have made a sincere attempt to eliminate such errors in the revised text.

In alpha-def-1 IV injection methodology, was 10 or 30 µg the total amount or is this µg/kg of bodyweight? 

10 or 30 µg are the total amount. The average weight of the mice was ~25 g. Thus, the dose of �-def-1 was 0.4 mg/kg (the low dose) and 1.2 mg/kg (the high dose).

This is human alpha-def-1? How much did the overall weight of the mice vary to account for the differences in relative amount of alpha-def-1 received by each individual across the treatment cohort. In addition, it is unclear if this is a tail vein injection.

Thank you for the opportunity to address these concerns. First, all studies were performed with humans α-def-1. The two sources of α-def-1 are specified in more detail in revised manuscript (Please page 4, Materials section lines 3- 5). Both proteins had the same effect formation of complexes with LDL in vitro and acceleration of LDL clearance in vivo (J Biol Chem. 2016; 29: 2777). Second, there was no significant difference in the weights of apoE-/- mice that did or did not receive α-def-1or cholestyramine. This has been added to the revised text (Please see page 4, Cholesterol-lowering section lines 6-7). Third, we now specify that the injections of alpha-def-1 were made by injection into the tail vein (Please see page 4, Cholesterol-lowering section lines 3-4).

Def++ mice need to be defined when introduced in the discussion, and rationale for using ApoE-/- mice be made more clear.

The mice mentioned in this paper are the progeny of those previously described in detail and reported by us. They are the same mice provided by one of us (KB) to Paulin et al. We have noted this is the revised Discussion (Please see page 6, line 2 in the Discussion).

Figure legends.

1A. Clarify H & L indicate high and low, this can be included in the key as there is space. Whether the last gray bar is MFD or MHFD needs to be resolved, this and the high/low can be applied to subsequent panels. Finally, what is not significant and what is significant as indicated by the p<0.001 in the middle of the figure?

The terms “H” and “L” have now been clarified in the legend to Figure 1A (Please see page 8, legend figure 1 A line 3-4). We now use the term MFD throughout. We have revised Figure 1A for clarity. There was no statistically significant difference between results in mice receiving a low dose of cholestyramine and a low dose of �-def-1. There were also no statistically significant differences between outcomes in mice receiving a high dose of cholestyramine, a high dose of α-def-1 or on MFD. Significant differences (p < 0.001) were found between the two former groups and the last three groups (now shown in the figure) and between the control HFD and all other groups (as stated in the text in Results).

1B. The red and green asterisk description at present is unclear whether there is significant differences between low does Cholest or alpha-def 1 or whether the high and low dose treatments were compared. The Y axis labelling needs to be more clearly described in the legend.

We revised Figure 1B for clarity. In doing so, we deleted the red and green asterisks. Lesion diameters in mice given a low dose of cholestyramine (Cholest L) were compared with those in mice given a low dose of �-def (�-Def L). Lesion diameters in mice given a high dose of cholestyramine (Cholest H) were compared with outcomes in mice given a high dose of �-def (� -Def H). Mice given �-Def H were compared to mice given a modified fat diet (MFD). In each case, the differences were statistically significant with a P value of <0.001.

1C. Figure legend needs to include this is representative aortic root sections... Also, there is no alpha-def high representative image.

We have corrected the error in the label. 

Is intimal damage score expressed as a percentage similar to Figure 1B. At present, the legend is difficult for the reader to interpret that data from Figure 1 is being analyzed and needs clarification.

We have revised the legend to state the section shown is representative of the 16 lesions analyzed in Panel B.

Overall, significant clarification in the discussion is needed. Clearly, there are multiple variables at play between alpha-def-1 and atheroprogression, especially in previous studies and the data presented in the current study, however the conclusions need to better match the data presented herein and outline what additional questions remain. First, this study attempts to control for the cholesterol-lowering effects of alpha-def-1 by including a treatment group is administered cholestyramine, and extends on previously published reports by including two different concentrations of alpha-def-1. It needs be abundantly clear in the text that while similar decreases in cholesterol were observed, alpha-def-1 treated mice had larger lesion size in comparison to cholestyramine treated mice. At present, what the authors are communicating by including the second to last paragraph of the discussion is unknown, however the correlation between the two studies overall shows a correlation between reductions in cholesterol and lesion size with the presence of alpha-def-1.

The Discussion has been revised to include the suggestions of the Reviewer (See page 7, 3nd paragraph lines 4-7 and last paragraph beginning in lines 6). 

There are several key questions, would combination of cholestyramine and alpha-def-1 increase the lesion volume (due to alpha-def-1 activities?)? 

As suggested by the Reviewer, we performed a pilot study in which we combined low doses of α-def-1 and low doses of cholestyramine in ApoE-/- mice on a high fat diet. Lesion size was larger in mice given both reagents than in mice given cholestyramine alone. We did not add this data to the manuscript because it does not change our conclusions but we are happy to do so. 

If alpha-def-1 was inhibited (or inactivated) in the present study, would that serve to increase both lesion size and cholesterol levels, or would lesion size be reduced in comparison to the ApoE-/- mice on the HFD, both in the presence or absence of cholestyramine? 

The Reviewer suggests what in theory would be an important experiment. Unfortunately, no antagonists of �-defs or means to inactive �-defs have been developed to the best of our knowledge. This is why in prior studies, we (1) compared results in HNP1-expressing transgenic mice with syngeneic wild type controls that do not express �-defs, (2) showed outcomes after bone marrow transplantation of �-def mice into wild type and wild type marrow into the transgenic, and (3) inhibited release of �-defs from neutrophils in vitro and in vivo with colchicine – all with the same result. Data using all three approaches show that decreasing plasma concentrations of �-defs tracks with smaller atherosclerotic changes in the aortic roots even though plasma LDL levels increase (J Biol Chem. 2016; 29: 2777). As underscored in the revised Discussion, these results are in line with recent clinical observations that colchicine reduces cardiac endpoints (N Engl J Med. 2019; 381: 2497) although it increase plasma concentrations of atherosclerotic lipoproteins (J Clin Lipidol. 2019; 13: 1016).

Would genetic modification to have Def+/+/ApoE-/- mice on a HFD result in mice with more atherogenesis that is reduced with something like colchicine, all these data would be helpful in supporting the authors current claims.

The Reviewer makes an interesting suggestion, but one that could take considerable effort and time to address and might or might not affect the central conclusions of our paper. In a simpler experimental system, Def+/+ mice on a regular or a high fat diet developed larger aortic lesions (J Biol Chem. 2016; 29: 2777) and colchicine reduced the effect (J Biol Chem. 2016; 29: 2777). A similar anti-atherogenic effect of colchicine was recently reported in humans (N Engl J Med. 2019; 381: 2497). Therefore, we would hypothesize that colchicine would reduce atherogenesis in Def+/+/ApoE-/- mice. On the other hand, double transgenic mice would create a less physiological system with extraordinarily high LDL and triglyceride levels and thus may generate novel interactions that are not relevant to physiological conditions and may be hard to interpret. Therefore, the proposed scheme might will not provide new compelling insights into our central thesis, i.e. that ��defs may contribute to the development of atherosclerosis in the clinical setting of normal levels of LDL. We hope the Reviewer will see this as a formidable and potentially unrevealing task. 

The data and previous studies show clearly that alpha-def-1 and neutrophils play a role in atherogenesis, one that might be a dual role, however the final sentence stating ‘inhibition of alpha-defs release or activity might help reduce the incidence or severity of atherosclerotic vascular disease and its thrombotic sequelae’ doesn’t appear to be supported currently by the data presented by the authors as in comparison to the control HFD, cholesterol and lesions are reduced.

We appreciate the Reviewer’s insight. We changed the sentence accordingly (Please see last 8 lines of Discussion section). As mentioned, in support of our hypothesis, newly published data by others show a dissociation between the rise in atherogenic LDL in patients on colchicine (J Clin Lipidol. 2019; 13: 1016) and a reduction in cardiovascular events (N Engl J Med. 2019; 381: 2497).

---

## [Decision Letter · Decision Letter 1]

20 Mar 2020

PONE-D-19-31479R1

Opposing effects of HNP1 (α-defensin-1) on plasma cholesterol and atherogenesis

PLOS ONE

Dear Prof. Higazi,

Thank you for submitting your manuscript to PLOS ONE. After careful consideration, we feel that it has merit but does not fully meet PLOS ONE’s publication criteria as it currently stands. Therefore, we invite you to submit a revised version of the manuscript that addresses the points still raised by reviewer 2.

We would appreciate receiving your revised manuscript by May 04 2020 11:59PM. To enhance the reproducibility of your results, we recommend that if applicable you deposit your laboratory protocols in protocols.io, where a protocol can be assigned its own identifier (DOI) such that it can be cited independently in the future. For instructions see: http://journals.plos.org/plosone/s/submission-guidelines#loc-laboratory-protocols

We look forward to receiving your revised manuscript.

Kind regards,

Michael Bader

Academic Editor

PLOS ONE

Reviewers' comments:

Reviewer's Responses to Questions

**Comments to the Author**

1. If the authors have adequately addressed your comments raised in a previous round of review and you feel that this manuscript is now acceptable for publication, you may indicate that here to bypass the “Comments to the Author” section, enter your conflict of interest statement in the “Confidential to Editor” section, and submit your "Accept" recommendation.

Reviewer #1: All comments have been addressed

Reviewer #2: All comments have been addressed

2. Is the manuscript technically sound, and do the data support the conclusions?

Reviewer #1: Partly

Reviewer #2: Yes

3. Has the statistical analysis been performed appropriately and rigorously? 

Reviewer #1: I Don't Know

Reviewer #2: Yes

4. Have the authors made all data underlying the findings in their manuscript fully available?

Reviewer #1: Yes

Reviewer #2: Yes

5. Is the manuscript presented in an intelligible fashion and written in standard English?

Reviewer #1: Yes

Reviewer #2: No

6. Review Comments to the Author

Reviewer #1: (No Response)

Reviewer #2: Higazi et al. have sufficiently revised their manuscript. There are several minor copyedits to be made by the authors, which hopefully will help. I apologize as a reviewer if I missed any others.

1. Second introduction paragraph – comma needed ‘… liver, leading to hypercholesterolemia.’ Following sentence also needs to be broken up, it is five lines long with only one comma.

2. Methods: correct to Sigma-Aldrich in materials.

3. 4th results paragraph – ‘However, a significantly higher proportion [of] the aortic roots circumferences…’ Add of please.

4. Last results paragraph, first sentence: ‘As [a] second approach…’ Add a please.

5. Figure 1 legend: Fix parentheses for Choles L and H, several closed parentheses appear to be missing.

7. PLOS authors have the option to publish the peer review history of their article (what does this mean?). If published, this will include your full peer review and any attached files.

Reviewer #1: No

Reviewer #2: No

---

## [Author Response · Author response to Decision Letter 1]

25 Mar 2020

Dr. Michael Bader

Academic Editor

PLOS ONE

Dear Dr. Bader:

We are writing in reference to MS: PONE-D-19-31479 “Opposing effects of HNP1 (α-defensin-1) on plasma cholesterol and atherogenesis” that we revised and are re-submitting to your journal.

We appreciate the efforts taken by the reviewers and the editor to examine our study in detail, consider it of merit for publication in PLOS ONE, and inviting us to submit a revised version of the manuscript.

In the revised version of our manuscript we addressed all the concerns raised by reviewer 2. We have corrected the errors noted by the reviewer and have made a sincere effort by a thorough re-reading to avoid any errors, including copyediting, in the revised text. 

We hope that the revised manuscript is now suitable for publication in PLOS ONE.

---

## [Editor Report · Decision Letter 2]

27 Mar 2020

Opposing effects of HNP1 (α-defensin-1) on plasma cholesterol and atherogenesis

PONE-D-19-31479R2

Dear Dr. Higazi,

We are pleased to inform you that your manuscript has been judged scientifically suitable for publication and will be formally accepted for publication once it complies with all outstanding technical requirements.

With kind regards,

Michael Bader

Academic Editor

PLOS ONE
---

## [Editor Report · Acceptance letter]

31 Mar 2020

PONE-D-19-31479R2 

Opposing effects of HNP1 (α-defensin-1) on plasma cholesterol and atherogenesis 

Dear Dr. Higazi:

I am pleased to inform you that your manuscript has been deemed suitable for publication in PLOS ONE. Congratulations! Your manuscript is now with our production department. 

With kind regards,

on behalf of

Prof. Michael Bader 

Academic Editor

PLOS ONE